# Diurnal and Seasonal Solar Induced Chlorophyll Fluorescence and Photosynthesis in a Boreal Scots Pine Canopy

**Caroline J. Nichol [1],\*, Guillaume Drolet [1,2], Albert Porcar-Castell [3], Tom Wade [1], Neus Sabater [4,5], Elizabeth M. Middleton [6], Chris MacLellan [7], Janne Levula [8], Ivan Mammarella [9], Timo Vesala [9] and Jon Atherton [3]**

[1]  School of GeoSciences, University of Edinburgh, Alexander Crum Brown Road, Edinburgh EH9 3FF, Scotland, UK; Guillaume.Drolet@mffp.gouv.qc.ca (G.D.); tom.wade@ed.ac.uk (T.W.)

[2]  Direction de la recherche forestière, Ministère des Forêts, de la Faune et des Parcs, 2700 rue Einstein, Québec, QC G1P 3W8, Canada

[3]  Optics of Photosynthesis Laboratory, Institute for Atmospheric and Earth System Research Forest Sciences, University of Helsinki, P.O. Box 27, 00014 Helsinki, Finland; joan.porcar@helsinki.fi (A.P.-C.); jon.atherton@helsinki.fi (J.A.)

[4]  Image Processing Laboratory (IPL), Parc Científic, Universitat de València, 46980 Paterna, València, Spain; neus.sabater@fmi.fi

[5]  Finnish Meteorological Institute, Erik Palmenin Aukio 1, P.O. Box 501, FI-00101 Helsinki, Finland

[6]  NASA Goddard Space Flight Center, Greenbelt, MD 20740, USA; elizabeth.m.middleton@nasa.gov

[7]  NERC Field Spectroscopy Facility, School of GeoSciences, Grant Institute, West Mains Road, Edinburgh EH9 3JW, UK; chris.maclellan@ed.ac.uk

[8]  Institute for Atmospheric and Earth System Research/Physics, Faculty of Science, University of Helsinki, Hyytiäläntie 124, FI-35500 Korkeakoski, Finland; janne.levula@helsinki.fi

[9]  Institute for Atmospheric and Earth System Research/Physics, Faculty of Science, University of Helsinki, PO Box 48, FI-00014 Helsinki, Finland; ivan.mammarella@helsinki.fi (I.M.); timo.vesala@helsinki.fi (T.V.)

\*  Correspondence: Caroline.Nichol@ed.ac.uk

**Abstract:** Solar induced chlorophyll fluorescence has been shown to be increasingly an useful proxy for the estimation of gross primary productivity (GPP), at a range of spatial scales. Here, we explore the seasonality in a continuous time series of canopy solar induced fluorescence (hereafter SiF) and its relation to canopy gross primary production (GPP), canopy light use efficiency (LUE), and direct estimates of leaf level photochemical efficiency in an evergreen canopy. SiF was calculated using infilling in two bands from the incoming and reflected radiance using a pair of Ocean Optics USB2000+ spectrometers operated in a dual field of view mode, sampling at a 30 min time step using custom written automated software, from early spring through until autumn in 2011. The optical system was mounted on a tower of 18 m height adjacent to an eddy covariance system, to observe a boreal forest ecosystem dominated by Scots pine. (*Pinus sylvestris*) A Walz MONITORING-PAM, multi fluorimeter system, was simultaneously mounted within the canopy adjacent to the footprint sampled by the optical system. Following correction of the SiF data for $O_2$ and structural effects, SiF, SiF yield, LUE, the photochemicsl reflectance index (PRI), and the normalized difference vegetation index (NDVI) exhibited a seasonal pattern that followed GPP sampled by the eddy covariance system. Due to the complexities of solar azimuth and zenith angle (SZA) over the season on the SiF signal, correlations between SiF, SiF yield, GPP, and LUE were assessed on SZA <50° and under strictly clear sky conditions. Correlations found, even under these screened scenarios, resulted around ~$r^2$ = 0.3. The diurnal responses of SiF, SiF yield, PAM estimates of effective quantum yield ($\Delta F/F_m'$), and meteorological parameters demonstrated some agreement over the diurnal cycle. The challenges inherent in SiF retrievals in boreal evergreen ecosystems are discussed.

**Keywords:** solar-induced chlorophyll fluorescence (SiF); seasonal dynamics; photosynthetic efficiency; proximal remote sensing; coniferous forest; gross primary productivity (GPP); light-use efficiency (LUE); Fraunhofer Line Discriminator (FLD); flux tower

---

## 1. Introduction

Photosynthesis, the light-driven conversion by plants of atmospheric carbon dioxide ($CO_2$) into carbohydrates, is one of the Earth's largest sinks for $CO_2$, thus playing a key role in determining the global carbon (C) balance [1,2]. The spatial and temporal dynamics of photosynthetic $CO_2$ uptake (Gross Primary Productivity, or GPP) can provide key information about the magnitude and variability of the fraction of atmospheric $CO_2$ that is absorbed by the terrestrial surface. Direct quantification of GPP can be carried out via micrometeorological methods, although this approach measures the Net Ecosystem Exchange (NEE), which is a combination of both GPP and ecosystem respiration, and therefore these processes need to be disentangled. NEE measurement, although being routinely made at hundreds of flux sites around the world, falls short in representing the spatial and temporal variability of these processes globally [3]. Current methods used for estimating GPP include: (1) A combination of eddy covariance, remote sensing and gridded satellite climate products, e.g., Reference [4], (2) satellite metrics of greenness and climate variables through modelling approaches [5], and (3) process-based models integrated into Earth system models [6]. However, in all these methods uncertainties propagate through to the final GPP estimates, and for this reason a need for independent estimates via direct observation still remains.

Improving our ability to monitor photosynthetic C uptake globally has become of major interest to climate and earth system modelers and to the remote sensing community, as this has direct implications for the development of climate mitigation scenarios or for determining targets in reduction of fossil fuel emissions (IPCC 2007). Currently, two main approaches exist to estimate photosynthesis directly from remotely sensed measurements. The first is based on measurements of the Photochemical Reflectance Index (PRI), a numerical index that uses the spectral reflectance values measured in a narrow detection band at 531 nm and in a reference band at 570 nm [7]. At short time scales (e.g., seconds, hours), variations in the PRI can be related to the amount of absorbed energy in the leaf that is directed to non-photochemical quenching (NPQ), a process by which plants under stress will dissipate excess absorbed energy as heat [8–10]. During that process, xanthophyll pigments in the chloroplast undergo reversible or sustained changes through de-epoxidation of violaxanthin into zeaxanthin, resulting in changes in leaf-level reflectance at 531 nm that can be related to canopy-level photosynthetic down-regulation. At longer time scales (e.g., days, months), variations in the PRI become increasingly related to changes in carotenoids and chlorophyll pigment pools [11,12], which have also been related to canopy photosynthetic light-use efficiency (LUE) [13,14]. The second approach for estimating photosynthesis from remotely sensed measurements is based on solar-induced chlorophyll fluorescence (SiF): The weak and near-instantaneous emission by chlorophyll molecules of redshifted light (i.e., lower energy relative to absorbed light) during photosynthesis [15]. In actively photosynthesizing plants, the amount SiF changes continuously in response to variations in environmental conditions.

While SiF is very small compared to the amount of energy that is reflected, it is possible to indirectly measure SiF through the application of appropriate algorithms such as those based on Fraunhofer Line Discriminator, Spectral fitting, or indirectly by using spectral indices based on the reflectance measured at SiF-related wavelengths [16–20]. Remote sensing of SiF is complex and challenging; apart from fluctuating in response to environmental conditions, the small amount of SiF also depends on the leaf or canopy chlorophyll content and on the fraction of energy that is directed towards NPQ relative to SiF and photochemistry [15,21,22]. As NPQ and SiF are both closely linked to photosynthetic efficiency, an increase/decrease in the amount of energy directed to any of them will

lead to a decrease/increase in the energy available for the remaining two photosynthetic pathways. Therefore, if we can accurately estimate SiF and NPQ, it is then possible to have a measure of the level of efficiency of carbon uptake by plants. While both the PRI and SiF-based approaches were successfully applied at the plant and small canopy levels [23–25], their application to larger spatial scales using airborne and satellite data remains challenging [26–34].

However, recent articles have highlighted, with varying degrees of success, the ability to extract SiF from orbiting platforms (e.g., from the Japanese Greenhouse gases Observing SATellite, GOSAT, and from the Global Ozone Monitoring Experiment-2, GOME-2) for estimating GPP [27,35–37]. These findings highlighted that spatial and temporal patterns of satellite retrieved SiF were highly correlated with GPP at the biome and global levels [33], but such correlations weaken at individual higher temporal and spatial resolutions [38] or during extreme climatic events [39]. Field based studies are continuing to emerge [20,25,29,40–44] though long term data sets, covering multiple seasons, are still rare [45–47]. Additionally, many of these studies have been restricted to crop or shrub canopies (e.g., References [47–50]) though one recent study has been in a deciduous forest [25].

In this study, we obtained a high frequency time series of canopy SiF, from early spring, during physiological inactivity, to peak physiological activity in summer, and decrease into autumn, over an evergreen boreal scots pine canopy, alongside measurements of PAM fluorescence, GPP from eddy covariance and environmental variables. We aimed to address the following questions: (1) How does SiF change over the growing season in a boreal evergreen ecosystem experiencing modest changes in APAR? (2) Is the SiF-GPP relationship as good as previously reported for crops and deciduous forests? (3) Is SiF able to capture changes in canopy physiological dynamics (LUE)?

## 2. Materials and Methods

### 2.1. Site Description

The study site was located at the Station for Measuring Forest Ecosystem-Atmosphere Relations (SMEAR II) in Hyytiälä, Southern Finland (61.847 N, 24.294 E, elevation 181 m), which has been measuring ecosystem fluxes (e.g., $CO_2$, $H_2O$), climatic and ecological variables continuously since 1995 [51]. The studied canopy was a Scots pine (*Pinus sylvestris* L.) stand established in 1962, with stem density, mean tree height and mean diameter at breast height (1.3 m) of 755 stems ha$^{-1}$, 16 m and 18 cm, respectively [52]. Four years prior to our study, all-sided leaf area index (LAI) of pine trees around the eddy covariance tower was estimated at 6.5 [53], which corresponds to a projected LAI of approximately 3. Ground vegetation around SMEAR II is composed mostly of the ericaceous shrubs *Calluna vulgaris* L., *Vaccinium vitis-idaea* L. and *V. myrtillus* L., and of the moss species *Dicranum undulatum*. The terrain at the site is relatively flat, with a thin soil (50–150 cm) classified as a Typic Haplocryod [51,54]. The mean annual temperature is 4.3°C and the mean annual precipitation is 590 mm [52].

### 2.2. Spectral Data

The dual-field-of-view optical system which used a pair of Ocean Optics USB2000+ spectrometers operated in dual field of view mode, sampling at a 30 min time step using custom written automated software, and fully described in Reference [55] was installed on the SMEAR II eddy covariance (EC) flux tower during in early April 2011. It operated continuously between April 23 and September 30 (days of year 113 to 273), acquiring simultaneous incoming solar irradiance and canopy-reflected radiance every 15 min.

The canopy-pointing optical fiber, with a FOV of 24.8° (yielding an off nadir instantaneous FOV of approximately 400 m$^2$ on the canopy) was attached to the tower at 7 m above the tree tops, with azimuth and viewing angles of 280° and 70° (relative to nadir), respectively. This configuration allowed us to measure about 400 m$^2$ of mostly sunlit pine crowns when the sun was at 35° degrees or more above the horizon (see Figure 5 in Reference [55]) minimizing the impact of ground cover

dynamics on APAR (e.g., snow vs green ground vegetation) and facilitating the investigation of the impact of canopy LUE dynamics on SiF. The system cosine-corrected optical fiber (CC-3, Ocean Optics Inc., Dunedin, FL, USA) was attached to the tip of a vertical pole at the top of the tower, sitting at about 1 m above the canopy-pointing fiber. This resulted in an almost completely unobstructed view of the sky; only a small part of the nearby SMEAR II tall tower was included in the fiber IFOV.

Intensity counts recorded by the spectrometers were converted to radiation fluxes, namely downwelling irradiance and upwelling radiance, using the procedure described in Reference [55]. Calibration coefficients used to calculate fluxes were obtained from an on-site calibration of the optical system prior to the start of the study period. Calculated radiation fluxes were also used to derive a spectral reflectance spectrum for each periodic acquisition by the optical system.

Diurnal variations in cloud cover contribute to making physiological interpretation of SiF and spectral reflectance data more difficult. Apart from the obvious direct effects of clouds decreasing the overall PPFD and subsequently SiF signal, clouds also increase the ratio of diffuse-to-direct radiation, which will change the radiation field within the canopy and the relative contribution of different canopy components to the measured SiF signal. To minimize potential cloud effects on the spectral data, we visually inspected time series of 30 min photosynthetically active photon flux density (PPFD), which was measured for the whole sampling period by a quantum sensor (Li-190SZ, LI-COR Inc., Lincoln, NE, USA) on the nearby SMEAR II radiation tower. We split the dataset into clear and cloudy observations through a visual inspection of the shape of PPFD diurnal patterns: We considered as clear spectral data points only those ones that occurred on a Gaussian response in PPFD over the diurnal course [56]. Finally, to minimize the impact of structural effects on spectral data, we calculated solar zenith angle (SZA) values for each spectral acquisition using the site geographic coordinates and date-time of measurement; these were used as filtering conditions in regression analyses.

### 2.3. SiF and Spectral Indexes

A number of methods are reported for the extraction of SiF using the FLD method and variants on this approach [57] The approach taken in this paper utilised two band FLD approaches due to the relatively coarse resolution of the spectral data (1 nm resolution), obtaining the exact position of the two shoulders of the absorption feature accurately was not possible, and thus the more popular 3FLD approach was not used. SiF values were extracted from the spectral data using [16] applied to the $O_2$-A absorption region of the electromagnetic spectrum (~755–765 nm). Damm [57] reported that all FLD approaches were similarly affected when coarser resolution data was utilised, and the two wavelength approaches tended to overestimate SiF. Accordingly, because the absolute SiF values reported in the present study may not be free from overestimation, we focus the analysis on the patterns of variation rather than absolute levels.

Wavelength matching between pairs of temporally concomitant spectra, a requirement of the FLD method, was achieved by interpolating down-welling irradiance spectra to the same wavelengths as those of reflected radiance spectra using locally-weighted polynomial regression. Then, for each pair of measurements, wavelengths of local minima between 758 nm and 762 nm were identified in both irradiance and radiance spectra and were used as absorption wavelengths for retrieving energy fluxes. FLD shoulder wavelengths were identified by visual inspection of irradiance and radiance spectra; a constant value at 757.5 nm was used for both spectra and all retrievals.

To account for energy re-absorption by molecular oxygen along the photon path between the sensor and the canopy, we applied the correction proposed in Reference [58]. Atmospheric transmittance values in the $O_2$-A region were calculated for each SiF observation using the HITRAN molecular spectroscopic database [59,60] along with air temperature and atmospheric pressure values measured at the flux tower. All data presented in this paper have therefore been corrected for atmospheric effects.

Last but not least, because the amount of energy re-emitted by plants as fluorescence is directly proportional to the amount of incoming radiation, the functional interpretation of the SiF signal (either

in terms of Fluorescence Yield or in terms of photosynthetic energy absorption or APAR) is not straightforward. Signal normalization by incoming PAR or indeed APAR is commonly a challenge because downwelling PAR radiation registered over the canopy does not necessarily reflect the actual PAR experienced by the foliage within the upwelling sensor FOV, due for example to diurnal changes in sun/shade fractions. In an attempt to control for variation in PAR directly within the sensors IFOV, we calculated a relative SiF yield ($SiF\ L_{650}{}^{-1}$), which is adimensional, by normalizing SiF observations by the reflected radiance measured in the red region of the spectrum (650 nm). In other words, diurnal and seasonal variations in $L_{650}$ were used as proxy of average incoming PAR within the scene. The usage of reflected radiance ($L_\lambda$) as a proxy of variations in incoming PAR within the IFOV is based on two assumptions (See Equation (1) below): (i) Constancy in canopy reflectance in wavelength $\lambda$ ($\rho_\lambda$), and (ii) proportionality between irradiance in the PAR region ($E_{PAR}$) compared to that at wavelength $\lambda$ ($E_\lambda$).

$$L_\lambda = E_\lambda\,\rho_\lambda = \ k\,E_{PAR}\,\rho_\lambda \tag{1}$$

We preliminarily selected 650nm band, but other wavelengths within PAR spectral domain could be equally useful. Given the common use however of a normalisation of SiF using APAR, we further computed SiF/APAR for purposes of comparison with the above normalisation scheme used in this study. APAR is described and calculated according to Equation (6).

## 2.4. Leaf-Level Chlorophyll Fluorescence

To understand seasonal changes in SiF values derived from the optical system, leaf-level fluorescence measurements were carried out on needles from three Scots pine trees located near the IFOV of the optical system. These measurements were made using a system of field fluorometers (MONITORING-PAM, Heinz Walz GmbH, Effeltrich, Germany) [61,62]. Fluorometers were installed in the top canopy and South-facing shoots across three different trees. Every 15–30 min, a saturating pulse was applied to each sampled branch and actual and maximal fluorescence yield from light-exposed leaves ($F_t$ and $F'_m$, respectively), PPFD at sample level ($PPFD_{SL}$, in µmol m$^{-2}$ s$^{-1}$) and leaf temperature (°C) were recorded (see Reference [61] for more details). Quantum yield (i.e., operating efficiency) of photosystem II (PSII) photochemistry ($\Delta F/F_m{}'$), an indicator of the acclimation of photosynthetic reactions to environmental conditions, was calculated for each measurement as:

$$\frac{\Delta F}{F'_m} = \Phi\mathrm{PSII} = \frac{(F'_m - F_t)}{F_m{}'} \tag{2}$$

Values of $\Delta F/F_m{}'$ from all three fluorometers were averaged together for each measurement time. To match averaged $\Delta F/F_m{}'$ values with the times of the SiF observations, we applied a simple weighted average interpolation method:

$$\Phi\mathrm{PSII}_t = \Phi\mathrm{PSII}_{t-1}\,w_{t-1} + \ \Phi\mathrm{PSII}_{t+1}\,w_{t+1} \tag{3}$$

where $\Phi\mathrm{PSII}_t$ is the quantum yield of photochemistry at time $t$ of a given SiF observation, $\Phi\mathrm{PSII}_{t-1}$ and $\Phi\mathrm{PSII}_{t+1}$ are $\Phi\mathrm{PSII}$ values at the closest point in time before and after time $t$, respectively, and $w_{t-1}$ and $w_{t+1}$ are the normalized distances from time $t$ and calculated as:

$$w_{t-1} = 1 - \frac{(t - t_{t-1})}{(t_{t+1} - t_{t-1})}\ ;\ w_{t+1} = 1 - \frac{(t_{t+1} - t)}{(t_{t+1} - t_{t-1})} \tag{4}$$

where $t-1$ and $t+1$ are the closest observations in time before and after time $t$, respectively. Finally, time series of $\Phi\mathrm{PSII}_t$ were visually inspected and outliers removed. $\Phi\mathrm{PSII}_t$ will be referred to as $\Delta F/F_m{}'$ in the remainder of the paper.

### 2.5. Gross Primary Production and Light-Use Efficiency

Time series of gross primary production (GPP, in μmol m$^{-2}$ s$^{-1}$) and light-use efficiency (LUE) were derived from $CO_2$ net ecosystem exchange (NEE) measured using the eddy covariance technique [9]. NEE was calculated on a half-hourly basis, according to [63] from high-frequency measurements of $CO_2$ concentration and vertical wind speed obtained from a closed-path infrared gas analyser (Li-6262, LI-COR, Inc., Lincoln, NE, USA) and a three dimensional ultrasonic anemometer (Solent 1012R, Gill Instruments Ltd., Lymington, UK), respectively. Partitioning into gross ecosystem productivity (GEP) and ecosystem respiration was done according to Reference [53]. In this study, we assumed tower GEP to be equivalent to GPP and only the later term is used in the paper. Half-hourly LUE values ($LUE_{30}$) were calculated as:

$$LUE_{30} = \frac{GPP_{30}}{PAR_{30}} \qquad (5)$$

where $GPP_{30}$ and $PAR_{30}$ are the 30-min mean values of GPP and photosynthetically active radiation (PAR). We further calculated that absorbed by the canopy (APAR), respectively. $APAR_{30}$ was calculated as:

$$APAR_{30} = PAR_{30} \times fPAR_{30} \qquad (6)$$

Reflected shortwave radiation measurements from a pyranometer (TP-3, Reeman) mounted on the radiation tower were used as a proxy for $PAR_{30}$ in $fPAR_{30}$ calculations:

$fPAR_{30} = 1 - SW_{r30} /SW_{i30}$, where $SW_{r30}$ are 30-min means of reflected shortwave radiation and $SW_{i30}$ are 30 min means of incoming shortwave radiance matched to the time of $PAR_{30}$ observations.

### 2.6. Environmental Variables

Environmental conditions during the measurement period were assessed using time series of meteorological variables from sensors located on and around the SMEAR II towers: air temperature (°C) at 17 m (PT100 RTD, Omega Engineering, Inc., Norwalk, CT, USA) and precipitation (F12P, Vaisala, Helsinki, Finland). Measurements from NDVI and PRI sensors (SKR 1800 2-channel sensor, Skye Instruments Ltd., Llandrindod Wells, UK) mounted at 31 m on the SMEAR II tall mast were used to assess the seasonal dynamics of the canopy structure and function, as well as in comparisons with SiF, $SiF\ L_{650}{}^{-1}$, $GPP_{30}$, and $LUE_{30}$.

### 2.7. Statistical Analyses

Variables calculated from spectral measurements were matched with corresponding 30 min averages of fluxes, meteorological and leaf-level fluorescence variables. To achieve this, only spectral observations for the 15th and 45th minute of each hour were retained since they both fall in the middle of an averaging period for flux and meteorological variables (i.e., from start to the 30th minute, and from the 30th to the 60th minute of each hour, respectively). The resulting dataset was used in regression analyses presented in the next section.

## 3. Results

### 3.1. Seasonal Patterns of Environmental and Physiological Variables

Daily photosynthetic photon flux density (PPFD), precipitation (including snow), and air temperature throughout the sampling period (DOY 113–273) are shown in Figure 1. From day 113 through to 130, air temperatures remained close to zero degrees, then started to increase until they reach a summer maximum of 29°C around day 160 (June 9), and then decreased towards the autumn. Maximum PPFD values during the summer period were around 1500–1600 μmol m$^{-2}$ s$^{-1}$. A week of almost cloud-free days stretched from June 5 (day 156) through to June 11 (days 156–162), coinciding with the warmest days of the summer. Precipitations as rain occurred during each month

of the sampling period, but were more abundant in September (164 mm, 19 days with rain $\geq$ 1 mm) than in any other months of the sampling period, for which monthly precipitations varied between 22 mm (April) and 96 mm (August), and the number of days with rain accumulation of at least 1 mm was between 6 (April) and 11 days (May and June).

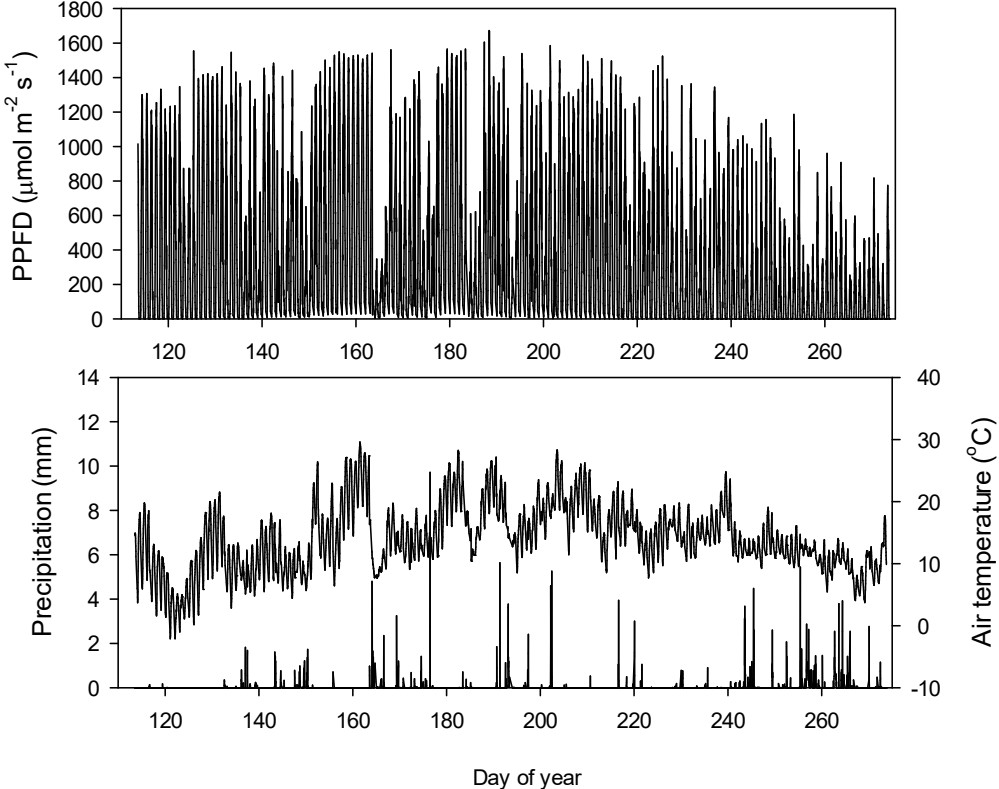

**Figure 1.** Seasonal course of meteorological conditions measured at the SMEAR II station during the sampling period. Top panel shows 30 min means of photosynthetic photon flux density (PPFD). The time series in the bottom panel shows 30 min means of air temperature measured at 17 m above the ground while vertical bars represent 30 min sums of precipitation.

Figure 2 shows samples of irradiance and reflected radiance spectra measured by the optical system on a clear day of the sampling period. The absorption feature related to the oxygen-A absorption band around 760 nm can be clearly appreciated in both curves (shaded area). In this region, the amount of energy reflected by the dark Scots pine canopy represents about 20% of the incoming energy, whereas in the visible region of the spectrum (400–700 nm), this reduces to between 3% and 5% only.

The seasonal course of SiF observations (day and night), including all ranges of solar zenith angles, diurnal variation, and without accounting for structural effects, light levels or cloud cover, exhibited a weak and noisy seasonal pattern throughout the sampling period (Figure 3A). After normalization by our proxy of incoming PAR in the IFOV (L650), $SiF\,L_{650}{}^{-1}$ presented a more consistent seasonal pattern, with a much smaller range of variation, and that more closely resembled that of $GPP_{30}$. The quantum yield of photochemistry measured by the PAM system also showed a typical seasonal pattern similar to those of $GPP_{30}$ and $SiF\,L_{650}{}^{-1}$, (Figure 3D). Note that the top envelope of points in Figure 3D represents night data and therefore corresponds to the widely used parameter $Fv/Fm$. Similarly, the lower envelope of points represents noon data points. An instrument malfunction resulted in a three week data gap in the $\Delta F/Fm\prime$ time series between July 26 and August 19 (days of year 207 to 231). Skye sensor PRI and NDVI both showed a slight incline through the measurement period into summer, although the seasonal pattern was flatter for PRI than NDVI. Skye sensor PRI and NDVI both showed a slight incline through the measurement period into summer, although the

seasonal pattern was flatter for PRI than NDVI. Importantly, the rapid increase in PRI towards day 130 (mid May) took place concomitantly with an increase in the apparent quantum yield of PSII (Figure 4G) and slight increase in LUE (Figure 4F), whereas $SiF \, L_{650}{}^{-1}$ remained stable during that period.

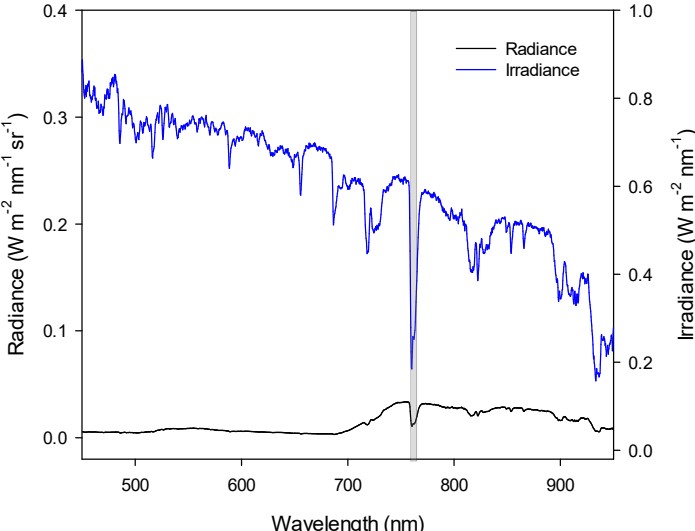

**Figure 2.** Sample irradiance and radiance spectra measured at 15-min intervals by the Ocean Optics spectrometer system used in this study. Each spectra is the average of 25 consecutive scans. The grey area shows the $O_2$-A absorption region around 760 nm used to extract SiF in this study. Note the difference between the ranges of the left and right Y axes, emphasizing the weak reflected signal from the Scots pine canopy.

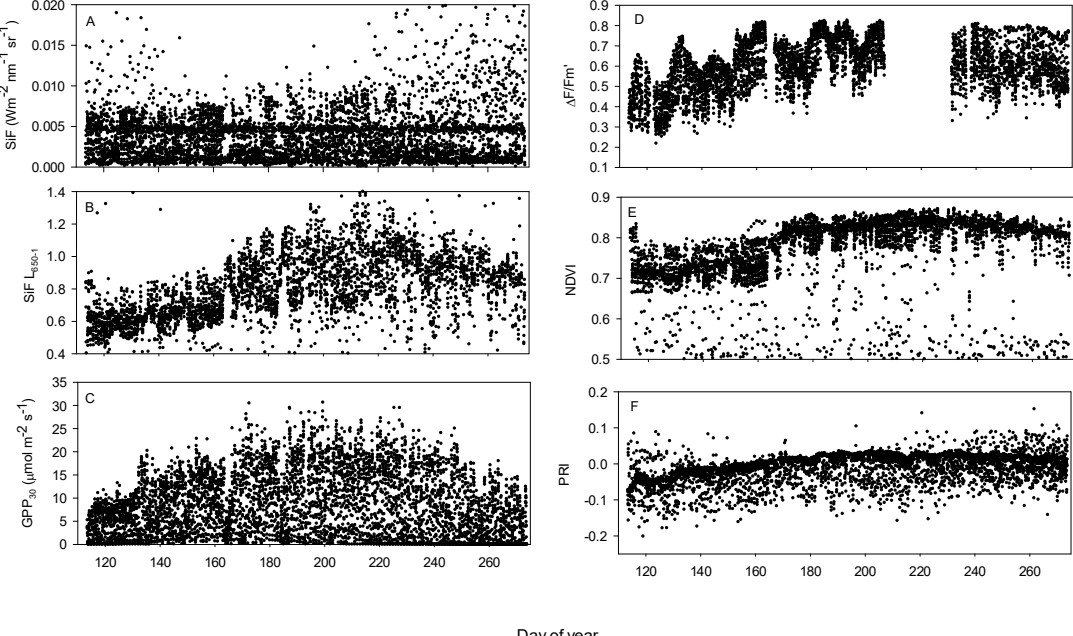

Day of year

**Figure 3.** Complete (day and night, all sky conditions) time series of instantaneous observations of (**A**) SiF and (**B**) $SiF \, L_{650}{}^{-1}$, both without correction for structural effects and centered on the flux data acquisitions, (**C**) 30 min means of gross primary productivity (GPP) calculated from the flux tower eddy covariance data, (**D**) effective quantum yield ($\Delta F / F_m{}'$) measured by the PAM system, (**E**) the normalized difference vegetation index (NDVI), and (**F**) the photochemical reflectance index (PRI), both measured by the Skye sensors mounted on the SMEAR II tall tower. A malfunction of the PAM system resulted in a gap in the $\Delta F / F_m{}'$ time series between days of the year 207 and 231.

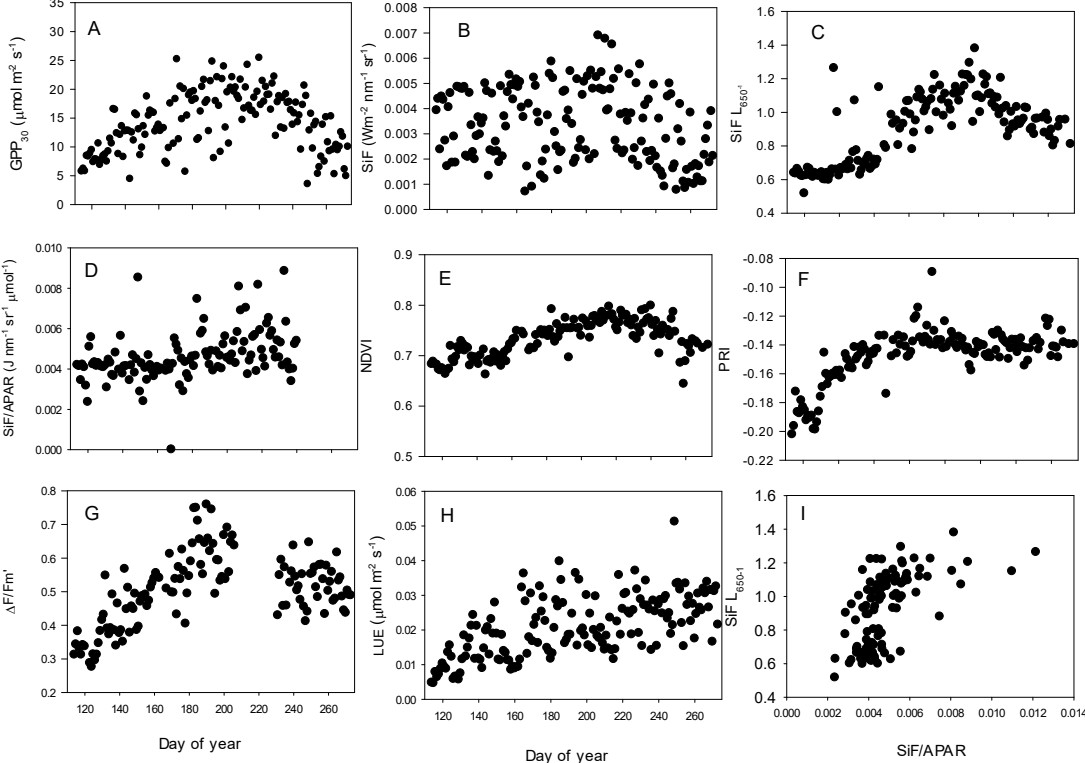

**Figure 4.** Seasonal patterns of a midday average (10–12 UTC) of: $GPP_{30}$ from flux tower data (**A**), SiF (**B**) and *SiF* $L_{650}{}^{-1}$ (**C**) from canopy level measurements from the Ocean optics spectrometers, SIF/APAR (**D**), PRI, (**E**) NDVI (**F**) PRI, and $\Delta F/F_m'$ (**G**) from the PAM system; LUE (**H**), (**I**) *SiF* $L_{650\text{-}1}$ to SiF/APAR. Rain and snow days were removed. Clear and cloudy observations are presented.

Seasonal patterns similar to those shown in Figure 3A–F, but with much less noise, were obtained by performing a midday average and removing observations from rain and snow days (Figure 4). Observations acquired during these periods are characterized by lower solar irradiance levels and thus, contribute to reducing the signal-to-noise ratio of the spectral data. They are also increasing noise levels in GPP estimates due to the small differences between NEE and ecosystem respiration fluxes around dawn and dusk. While accounting for low irradiance observations removed noise in the SiF time series, it did not improve its overall seasonality (Figure 4B). The seasonality in *SiF* $L_{650}{}^{-1}$ improved (Figure 4C) when the SiF signal was normalized by the radiance at 650 nm, but not when compared to the SiF normalized to APAR.

### 3.2. Relationships between SiF, SiF $L_{650}{}^{-1}$, $GPP_{30}$ and $LUE_{30}$

To explore the relationships between *SiF*, *SiF* $L_{650}{}^{-1}$ with $GPP_{30}$ and $LUE_{30}$, three steps were undertaken. Firstly all data were included in the analysis; these also included cloudy and clear observations as well as all solar angles (and shown in grey circles in Figure 5A–H). In a second step, data points were filtered to remove all observations that were not acquired on completely clear days (dark circles in Figure 5A,C,E,G). Finally, data points were filtered to also remove SZA values larger than 50°, thus reducing spectral data variations due to changing illuminated/shadowed canopy fractions (Figure 5B,D,F,H). The relationships between *SiF*, *SiF* $L_{650}{}^{-1}$ with $GPP_{30}$ and $LUE_{30}$, when considering data collected at all SZAs showed scattered and non-significant relationships (Figure 5A,C,E,G). There was marginal improvement in the relationships when data were filtered to include only higher SZAs (>50°), as shown in Figure 5B,D,F,H. Linear regression analyses applied to these data showed only weak correlations between the analyzed variables ($R^2$ 0.24–29, $p < 0.0001$) (Table 1).

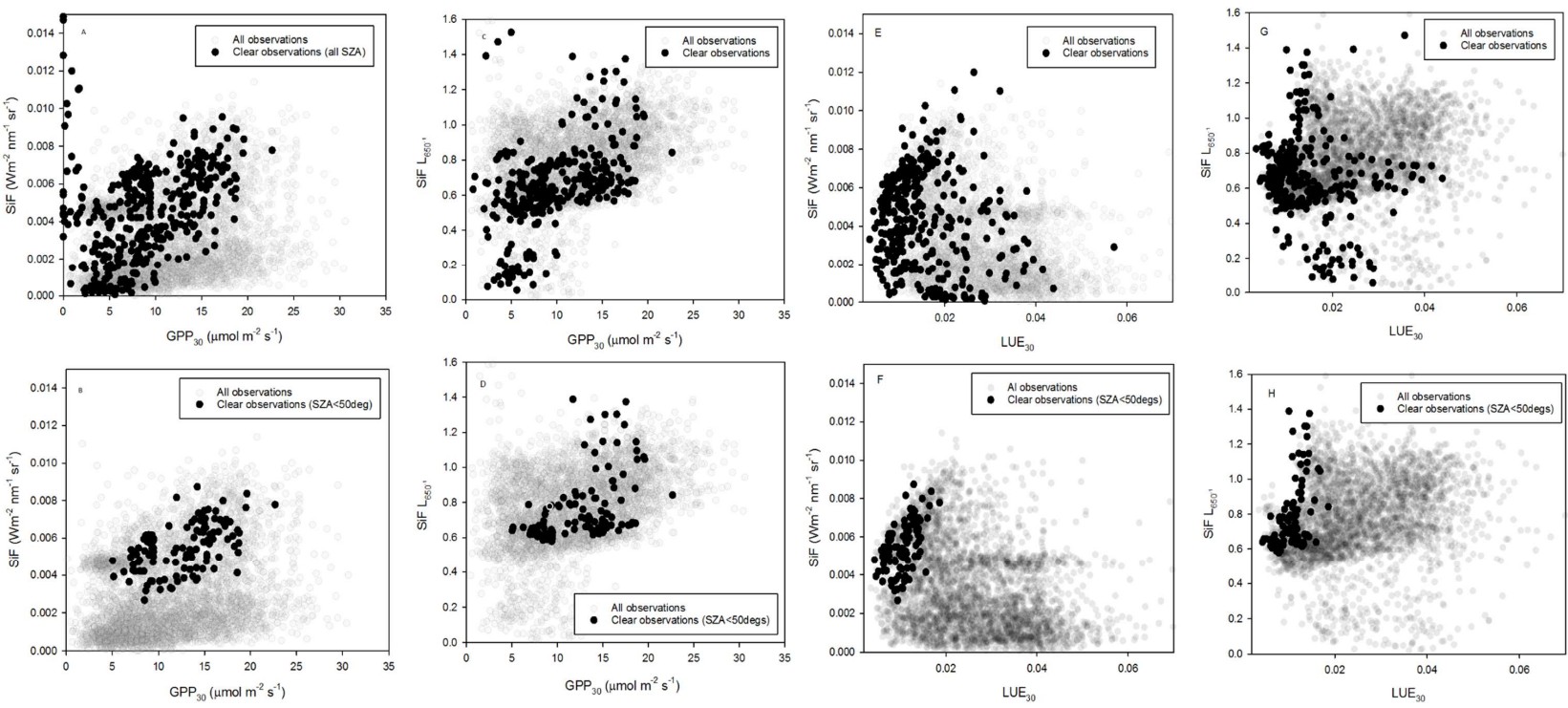

**Figure 5.** Relationships between SiF and *SiF* $L_{650}{}^{-1}$, and *GPP*$_{30}$ and *LUE*$_{30}$ (10–12 UTC). Grey circles include data acquired on clear and cloudy days as well as all solar angles. Black circles represent data acquired during clear days (**A,C,E,G**), and also when solar zenith angle was smaller than 50° (n = 118 for **B,D**, 116 for **F,H**) on clear days (**B,D,F,H**).

**Table 1.** Parameter estimates, R-square values and significance levels from linear regressions for the four lower panels in Figure 5.

|   | Slope | Intercept | R2 | Std Error | F | df | *p*-Value |
|---|---|---|---|---|---|---|---|
| B | 0.0002 | 0.0003 | 0.26 | 0.001 | 39.76 | 117 | <0.0001 |
| D | 0.0251 | 0.4468 | 0.25 | 0.166 | 38.15 | 117 | <0.0001 |
| F | 0.2123 | 0.0032 | 0.24 | 0.001 | 37.18 | 115 | <0.0001 |
| H | 34.6213 | 0.4091 | 0.29 | 0.163 | 44.72 | 115 | <0.0001 |

Interestingly, the relationship between SiF and GPP was better than between $SiF\ L_{650}{}^{-1}$ and GPP, which could be expected since both SIF and GPP are strongly controlled by PAR, whereas $SiF\ L_{650}{}^{-1}$ is readily normalized by PAR via $SiF\ L_{650}{}^{-1}$. In contrast, LUE was slightly better related to $SiF\ L_{650}{}^{-1}$ than to SiF, which was also expected since both LUE and $SiF\ L_{650}{}^{-1}$ undergo normalization by PAR.

We selected one clear day from each month from April through September and explored the relationships between SiF, GPP, PPFD, and SZA over a diurnal time course (Figure 7). GPP typically followed PPFD throughout the day in all days across the months, although its magnitude was smaller in the spring (April and May) than later during the growing season. During the early hours following sunrise, SiF increased following GPP and PPFD and peeked around 6 UTC. After this point, SiF decreased throughout the day, while GPP remained relatively stable or continued to increase until it peaked around solar noon. This pattern in SiF was consistent across clear days, but showed more variations on days when PPFD was less stable (July and September). Maximum SiF values occurred in May and July. Note that the time in Figure 7 is UTC, with solar noon occurring at 10AM.

Using the same clear days used in Figure 7, the diurnal relationships between leaf-level effective quantum yield from the PAM data, $SiF\ L_{650}{}^{-1}$ and SZA were also explored (Figure 6). During clear summer days, $SiF\ L_{650}{}^{-1}$ consistently showed a small peak around 4–5 UTC (Figure 6B–E). It then remained constant until around noon, when it started to increase until reaching a second peak of the same magnitude as that observed in the morning and which occurred from late- to mid-afternoon as the summer progressed. Past that afternoon peak, $SiF\ L_{650}{}^{-1}$ usually dropped rapidly toward the evening. PAM $\Delta F/F_m{}'$ showed diurnal courses going in opposite direction to those of $SiF\ L_{650}{}^{-1}$. Most months showed a maximum $\Delta F/F_m{}'$ in the early hours of the morning, which decreased until it reached a minimum value at a time that occurred increasingly later as the season progressed. Generally, $\Delta F/F_m{}'$ and $SiF\ L_{650}{}^{-1}$ both decreased from the morning until midday to early afternoon, after which they either moved in the same direction (Figure 6B,C) or diverged (Figure 6D,E). However, the time at which each variables reached a minimum daily value and started to increase and move in opposite directions as the season progressed. For $SiF\ L_{650}{}^{-1}$, this time occurred earlier during the day from April to September, while for $\Delta F/F_m{}'$ it occurred later as the months passed.

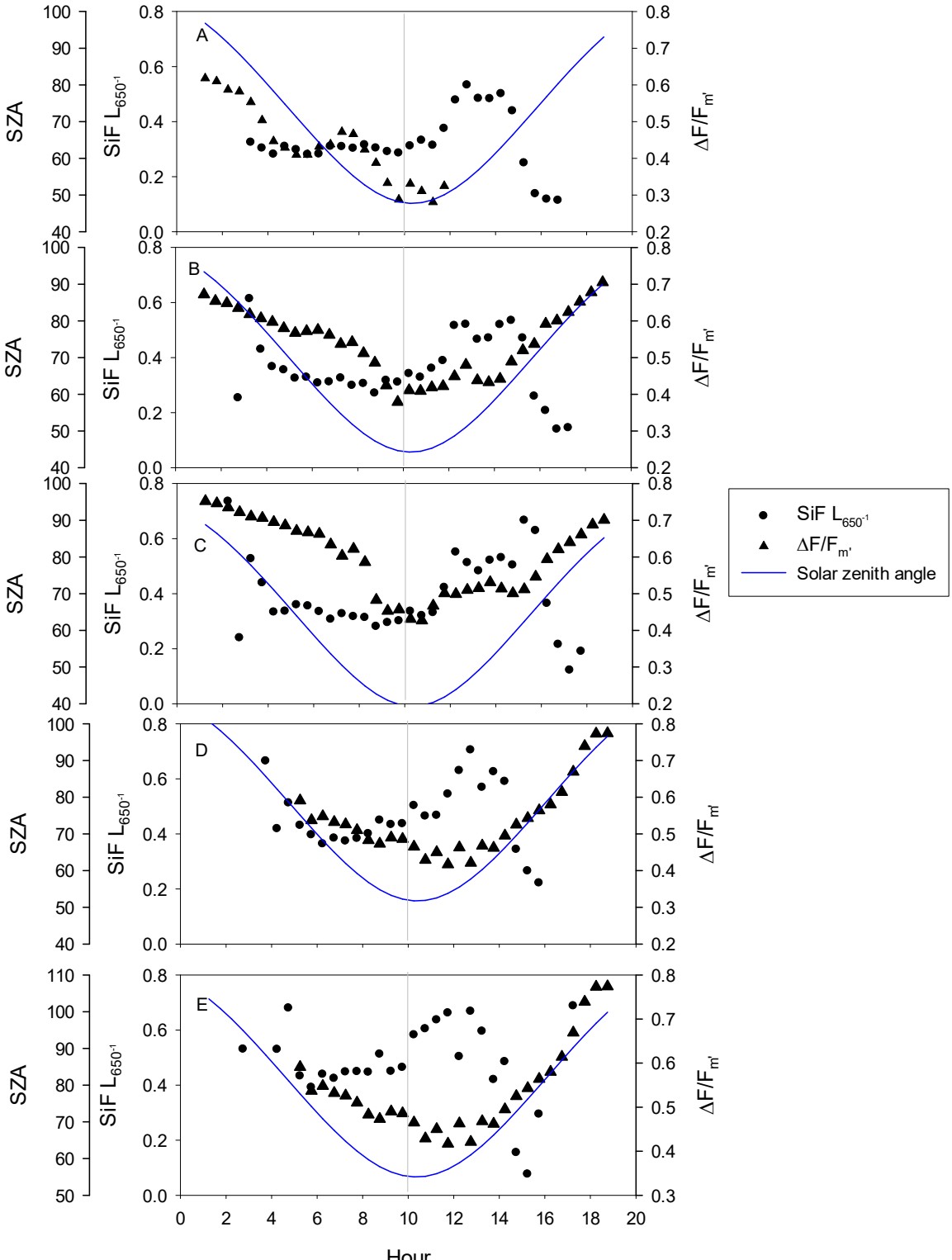

**Figure 6.** Diurnal course of *SiF* $L_{650}{}^{-1}$ (black circles), quantum yield of PSII ($\Delta F/F_m'$, black triangles) and solar zenith angle (SZA, blue curve) for the same clear days shown in Figure 7. (**A**) April 28th, (**B**) May 10th, (**C**) June 5th, (**D**) August 27th, and (**E**) September 5th. Note that there are no PAM data for July. Solar noon indicated by the grey line.

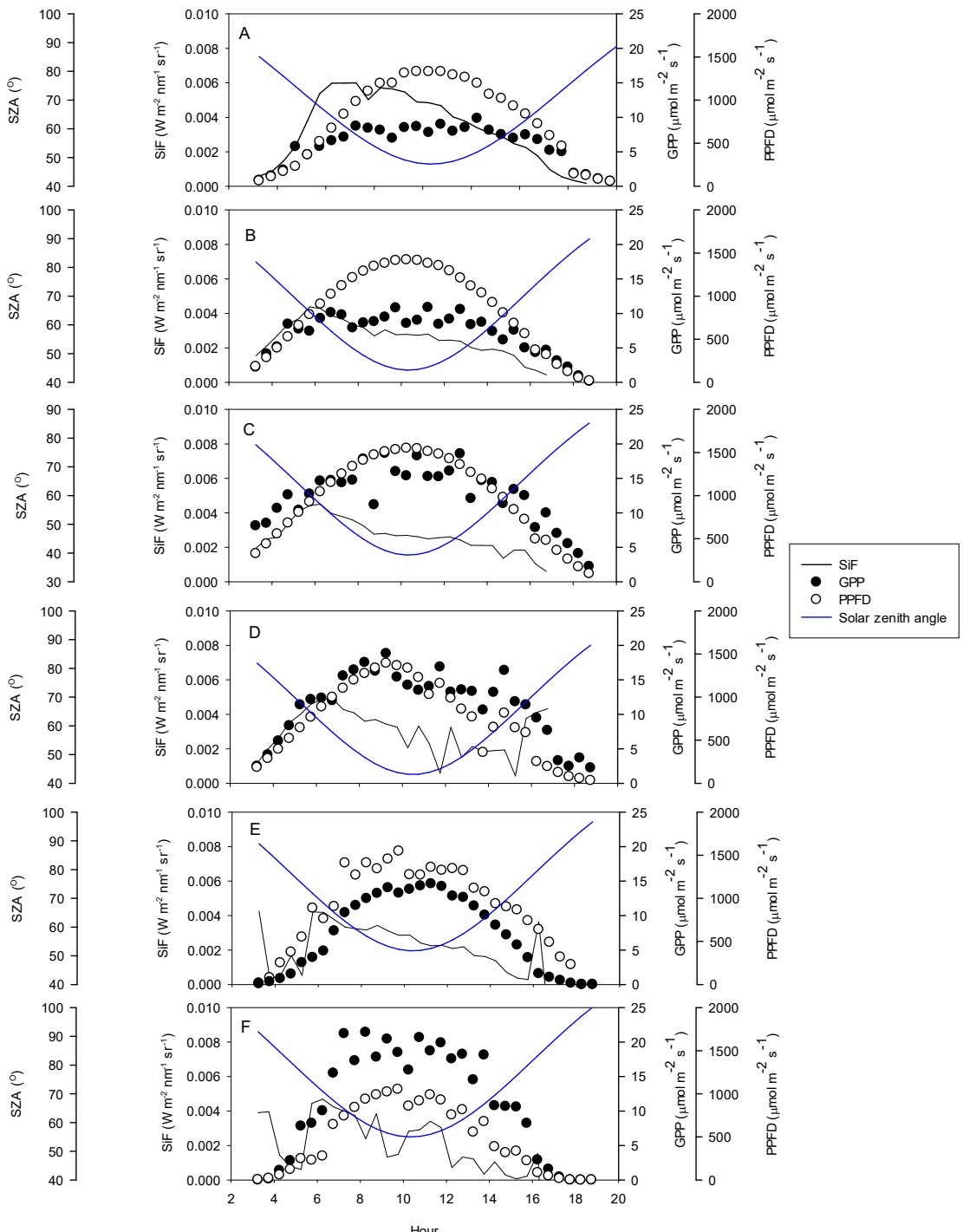

**Figure 7.** Diurnal relationship in SiF (-), GPP (solid circles), PPFD (open circles) and SZA (-) on select clear days throughout the growing season. (**A**) April 28th, (**B**) May 10th, (**C**) June 5th, (**D**) July 29th, (**E**) August 27th, and (**F**) September 5th.

## 4. Discussion

In this study, we deployed a tower-based optical system [55] to collect a continuous time series of canopy SiF over an evergreen canopy alongside measurements of PAM fluorescence, GPP from eddy covariance and environmental variables, NDVI and PRI, with the aim of describing both the diurnal and seasonal patterns of SiF in connection with ecosystem GPP dynamics. We aimed at exploring the following questions: (1) How does SiF change over the growing season in a boreal evergreen ecosystem

experiencing modest changes in APAR? (2) Is the SiF-GPP relationship as good as previously reported for crops and deciduous forests? (3) Is SiF able to capture changes in canopy physiological dynamics (LUE)? To the best of our knowledge, this is the first long-term time-series of SiF in an evergreen canopy from field-based instruments.

While a strong SiF-GPP relationship has been repeatedly observed at the ecosystem level using ground based instruments in crop and deciduous canopy types, [18,23–25,29,40–42,45,47,50,64–67], as well as at landscape/regional level from satellite SiF retrievals [26,27,31,33,34,68–73], the factors that drive these relationships remain controversial. Biome dependent differences in canopy structure have been suggested to affect the slope of the relationship [35,64,68,74], but recent work at the satellite pixel scale is suggestive of a universal relationship [31]. Similarly, the relative role that ecosystem APAR and LUE dynamics exert on linking SiF to GPP is also expected to be biome dependent, but its characterization is still limited. Observations across crops and deciduous forests suggest that APAR dynamics could be the main factor connecting SiF to GPP [25,73,75]. This is logical, since annual GPP dynamics for these vegetation types are dominated by APAR, which in turn controls both the SIF emission and photosynthetic $CO_2$ assimilation. The question remains as to how well SIF and GPP are related for an evergreen ecosystem, where only modest changes in APAR occur.

*4.1. Comparison between Evergreen Forest SiF and Other Terrestrial Ecosystems*

The seasonal midday signal of SiF, without any structural correction, was generally very noisy and did not exhibit any seasonal pattern (Figure 4B), this is in contrast to previous studies (albeit over crops and deciduous forests) that demonstrated a strongly seasonal pattern in SiF, [18,23–25,29,40–42,45,47,50,64–67]. We found only a very modest seasonal increase in NDVI (Figure 4E), as recorded by an independent broadband sensor installed 30 m above the canopy, likely indicative of minor changes in ecosystem APAR. In addition, because our upwelling radiation sensor was purposefully pointing towards the tree crowns, we would expect that APAR changes in the IFOV would be even smaller due to absent contribution from the ground vegetation. Taken together, these results indicate that the variation in absolute SiF of our evergreen pine canopy was dominated by incoming PAR. Normalization of SiF by incoming PAR has previously been carried out using PAR data acquired above the canopy [42], but unfortunately because of the dynamic sun/shade patterns within the IFOV during the course of the day and passing of the seasons, the actual PAR received by the foliage under examination will differ from that recorded above the canopy following complex temporal patterns and partly undermining the normalization. In an attempt to overcome this limitation, we used reflected radiance at 650 nm ($L_{650}$, see Equation (1)) as a proxy of actual PAR received by the foliage within the IFOV. After normalization by $L_{650}$, a clear seasonal pattern in *SiF* $L_{650}^{-1}$ emerged (Figure 4C), which more closely followed the time course of GPP (Figure 4A). The *SiF* $L_{650}^{-1}$ and indeed GPP, increased through the spring transition towards summer then declined into the autumn period. It is important to note that $L_{650}$ normalizes the signal by PAR, but not APAR, and therefore, increase in *SiF* $L_{650}^{-1}$ could still be due to both changes in APAR (e.g., new needle cohort and shoot elongation during June, and subsequent senescence of old needle cohorts during September), as well as leaf-level adjustments if fluorescence quantum yield (related to LUE). It has been demonstrated that normalisation (using various means such as PAR and APAR) can improve the SiF-GPP relationship in some cases [76], but it is recognised that it is not always possible due to unavailability of measurements like APAR. Certainly attention to various normalisation schemes is warranted in order to understand and separate the physiological and non-physiological components.

Our correlation analysis of *SiF*, *SiF* $L_{650}^{-1}$, and *GPP*$_{30}$, *LUE*$_{30}$ resulted in only weak relationships (Figure 5), which were again generally weaker than the SiF-GPP relationships reported in earlier seasonal and canopy-level studies for crops and deciduous forests or at larger scales using satellite data [35,43,47,50,73,77,78]. This demonstrated the complexity of extracting the physiological signal component from tower based SiF studies in complex forest ecosystems. To date few studies have focused on evergreen ecosystems. Walther [79], using SiF retrieved from the GOME-2 instrument,

explored the seasonality in SiF in a high latitude evergreen ecosystem, and reported a strong seasonal relationship between satellite SiF and modelled GPP. Furthermore, Wolfhart [39] studied a short time series (10 days in length) of SiF in an evergreen canopy that had experienced a short but intense heatwave, and found that during a short period of unchanging APAR, SiF was only very weakly related to the change in GPP, a change attributed to the fact APAR was almost unchanged during the analysis period. In our study, statistically significant relationships only emerged following filtering to limit analyses to midday, cloud free days with high SZA. The correction of SiF to $SiF\ L_{650}{}^{-1}$ improved the strength of the relationship with LUE but not with GPP (Table 1). This is logical since both SiF and GPP are strongly controlled by incoming PAR (indeed APAR), in contrast, $SiF\ L_{650}{}^{-1}$ is intrinsically normalized by PAR via $L_{650}{}^{-1}$. In fact, because $SiF\ L_{650}{}^{-1}$ is normalized by PAR, it becomes a relative measure of fluorescence yield and therefore should be expected to reflect the seasonal variations in photosynthetic LUE much better than SiF. Similarly long-term SiF (and SiF yield) measurements in a deciduous forest site at Harvard Forest [25] highlighted stronger relationships between SiF, SiF Yield, GPP and LUE, with similarly strong results between SiF and GPP reported in a mixed forest site [64]. Interestingly, the slight increase in apparent quantum yields of photochemistry (Figure 4G) and LUE (Figure 4H) observed here during early May, was registered by the PRI (Figure 4F) but not by $SiF\ L_{650}{}^{-1}$. Although this observation will require further validation and studies with even broader temporal coverage it preliminary points to potential limitations of SiF to track LUE in evergreens.

The nature of the relationship between SiF and GPP, continues to be questioned. Is it solely driven by the dependence on APAR or a combination of APAR and the photosynthetic light and dark-reactions? [66,78,80,81]. Long term and in situ measurement will prove particularly attractive in answering this question. Previous studies have highlighted that heat dissipation (NPQ) is the main driver of variations in fluorescence and photosystem yields, and the correlation between SiF yield and LUE is consistent with field studies and model-based assessments [25,82]. In the present study however, we found only a very weak correlation between SiF and GPP, and $SiF\ L_{650}{}^{-1}$ and LUE for the canopy component of an evergreen Scots pine forest.

### 4.2. Diurnal Relationships and Sun-Sensor Geometry

The diurnal patterns presented in this study (Figures 6 and 7) are likely to be controlled by both physiological and optical (directional) factors. Although the tower mounted optical system used in this study was oriented in a so called "hot spot" region of the canopy, when the solar angle varies through the day, so too will the fractions of sunlit and shaded foliage in the sensors field of view. The influence of directionality will therefore inevitably generate differing proportion of SiF coming from the scene viewed by the spectrometer. Understanding and decoupling the influence of both structure and physiological response from SiF retrievals is a much needed next step, and one that models have thus far been used to understand [82–84]. The passage of clouds may add orders of magnitude difference to the retrieved SiF (Figure 7D) via a significant change in the measured irradiance and subsequent depth of the $O_2$-A band [26]. In the results presented here the peaks of SiF and PPFD do not match, which is likely due to be dominated by the changes in solar geometry. Indeed the hourly fraction of sunlit and shaded leaves within the sensors FOV will be varying, as will the presence of multiple scattering within the canopy. Future work should undoubtedly focus on such shadow fraction changes, along with the impacts on SiF, in this ecosystem.

The influence of sun-sensor geometry on solar induced fluorescence itself is receiving increasing attention [85,86] with evidence from satellite retrieval analysis and indeed modelling studies mounting. The have reported noticeable angular influences each generating differing ranges of SiF values, which is similar to the effect of sun-sensor geometry on canopy reflectance data [65,84]. A bowl shaped response has been reported from backward to forward scattering directions consistently in ground and model based observation [65,84] (though not evergreen canopy specific). While it is not possible to disentangle the impact of sun-sensor geometry explicitly here, its influence remains highly probable.

*4.3. Atmospheric Influence and Instrument Resolution*

Retrievals of SiF in the oxygen absorption lines, from aircraft platforms hundreds of meters above the surface, have routinely been atmospherically corrected [48,64,87,88]. However, there has been little consensus until recently as to whether this correction would be needed for SiF in a proximal sensing context, i.e., from a flux tower mounted system within 20 m from the canopy. Historically SiF retrievals were formulated to be applied to top of canopy data, with (1) the assumptions of atmospheric path length between target and sensor is short enough to be neglected, and (2) that the solar irradiance is measured at the same height as the target. When atmospheric path length increases, for example with an instrument mounted on a tower or where large solar zenith and/or view angles are common, these assumptions cannot be met. Oxygen absorption is proportional to air pressure [89] and thus at this lowest level of the atmosphere even a few meters difference can result in significant error in retrieved SiF. In this study, oxygen transmittance was computed by making use of the HITRAN molecular spectroscopic database, therefore modelling the oxygen transmittance according to the experimental site configuration, e.g., fixing the optical path between the canopy and the sensor, and taking into account variations in the oxygen transmittance caused by changes in the environmental conditions, mainly temperature and pressure. Accounting for variations in the environmental conditions as part of the SiF retrieval strategy guarantees, especially in experimental sites subjected to abrupt changes of temperature and pressure, an accurate observation of the SiF seasonal patterns, as pointed out in Reference [58].

The complexity of the influences of instrument spectral resolution, signal-to-noise, atmospheric correction, canopy structure, leaf biochemical parameters and directional effects all play a critical role in shaping the reliability of SiF to quantify GPP [58,90,91]. Only recently have intercomparisons in SiF retrievals been carried out across optical instruments. Julitta [90] carried out tests on four spectrometers and investigated their ability to retrieve both red and far-red SiF. The work presented highlighted that an "accurate" far-red SiF could be retrieved from spectrometers with an ultra-fine resolution (less than 1 nm) with the red SiF estimation requiring a significantly higher resolution (less than 0.5 nm). The Ocean Optics (USB-2000) system used in this study had a 1 nm resolution, and would therefore fall at the upper end of the recommended resolution and thus may have hampered the SiF retrieval.

## 5. Conclusions

By linking high temporal resolution measurements of solar induced fluorescence with gross primary productivity, canopy light use efficiency and pulse amplitude measure of efficiency, we were able to, for the first time, explore the complex nature of understanding both SiF and its use in understanding evergreen canopy physiological properties. The results of study indicated that SiF alone did not prove useful over the seasonal cycle, and only after a correction term was applied to account for variation in illumination conditions, did the findings elucidate a seasonal cycle that mirrored GPP. While only a simple two-band FLD approach was utilized here, credible seasonal and diurnal trends were not dissimilar to published studies, though we highlighted the challenges in SiF retrieval in an evergreen canopy with a low LAI and low solar angles. The diurnal responses when compared to conventional PAM fluorescence highlighted structural and solar influence, with the nature of the diurnal relationship varying throughout the season. Further studies should invariably focus on higher resolution optical data and a longer time series. Only now are published works becoming available that highlight the very minimum in instrument setup and response, but no review of instruments, resolution, and methodology exists at the time of writing.

**Author Contributions:** C.J.N. was Principal Investigator for the project and carried out all analysis of the presented data. G.D. collected the Ocean Optics data and carried out its post processing. T.W., with input from C.M., built the Ocean Optics optical system and performed system checks before its use in this study. A.P.-C. and J.M.A. collected the PAM data and oversaw its post processing. A.P.-C. further contributed to the discussion of the results and instigated the use of the reflected radiance normalization idea. J.L. oversaw the collection of the eddy

covariance data and I.M. carried out its post processing. The manuscript was written by C.J.N. with significant written contributions from G.D., A.P-C., T.W., N.S., E.M., C.M., J.L., I.M., T.V., J.A.

**Funding:** This work was funded by the UK Natural Environmental Research Council NERC (NE/F01749), Finnish Academy (#293443, #288039), Academy of Finland Center of Excellence (projects No. 272041 and 118780) and ICOS-Finland (project No. 281255).

**Acknowledgments:** We thank Sergio Cogliati and Petya Campbell for their helpful discussions of the results presented in this paper.

**Conflicts of Interest:** The authors declare no conflict of interest.

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
