# Peer review of "Diurnal and Seasonal Solar Induced Chlorophyll Fluorescence and Photosynthesis in a Boreal Scots Pine Canopy"

_remotesensing, doi:10.3390/rs11030273_

Round 1

Reviewer 1 Report

In this study, the authors through measuring a high frequency time series of canopy SiF, from early spring to autumn. They discussed SiF change over the growing season in a boreal evergreen ecosystem experiencing modest changes in APAR, the SiF-GPP relationship and the ability of SiF to capture changes in canopy physiological dynamics (LUE). My comments on this manuscript are as given below:

1、The logic of the abstract is not clear and needs to be further improved, especially the main topic of this study (line 27-31).

2、Line 157: “we considered as clear spectral data points only those ones that occurred on a Gaussian response in PPFD over the diurnal course.” Please give reliable reference and explanation to this statement.

3、4.1. Comparison between Evergreen Forest SiF and Other Terrestrial Ecosystems, this part is mainly to explain the second question in this study, maybe can give a clear contrast chart which will be easy to understand for readers.

4、The conclusions need to be rewritten

5、In the discussion, the first and third questions were not clearly analyzed and summarized.

6、2.8 statistical analyses need to be more detailed.

7、The abscissa of Figure 6 need to be revised to “hours”

8、SIF and SiF need to be explained the differences between them.

9、How many is the amount of data about the smaller than 50° on clear days should be given.

10、   The number of significant digits of data of Table 2 should be more than four. The linear regression analysis for B, the slope and intercept become to zeros.

Author Response

Please find attached our document with thanks for the detailed and helpful comments. All are responded to in the attachment.

Kind regards

Caroline

Reviewer 2 Report

The authors presented the nice seasonal and diurnal variations in Solar-induced chlorophyll fluorescence measurements with other meteorological data. The lack of such measurements in the boreal forest ecosystems has been making it difficult to interpret the satellite-based Sif in the high latitudes. The author’s work can add the ground truth information of how the remote sensing index SIF varies seasonally diurnally. The data shown here is a very unique and thus the paper can be worth publishing in the remote sensing.

There is one general comment on the manuscript.

Section 4.2 (line 456 to 465): I agree that it is inevitable that Sif includes the directional effect like bidirectional reflectance factor. I want the authors to make a little more in-depth discussion along with the figure 6. According to the figure 6 the maximum sif occurs roughly four hours earlier than the local solar noon. The peaks of SIF and PPFR do not match so we can suspect that this mismatch is due to the geometry effect of the sensor pointing direction. If this is the case, there are two potential reasons: (1) hourly changes in the fraction of sunlit/shaded leaves and (2) multiple scattering of SIF within the canopy, which is affected by the hourly changes in sza. How do those two (or there might have other reasons) effect potentially contribute to the observed SiF? The former impact may be evaluated if the authors calculate the hourly change in fraction of sunlit leaves.

 Other comments

Line 35: please add the scientific name after common name Scots pine

Line 36: Provide the full name of PAM

Line 39: Provide the full name of PRI, NDVI

Line 61: “the fusion of a combination of eddy covariance, remote sensing and gridded satellite climate products” sounds odd because remote sensing and satellite products are almost equally.

Line 104: Please add the citation Colombo 2018.

Line 135: azimuth angle of 280-deg. Is this from north and clockwise? If so the sensor oriented west. Also, viewing angle of 70-deg from nadir seems nearly horizontal direction. Is this correct? What is the field of view of the canopy pointing fiber?

Line 155: Did authors split the data visually or apply a statistical approach such as the criteria based on the Gaussian fitting?

In Section 2.1: There is nothing about the characteristics of spectrometer used in this study and their field deployment information.

Line 198: “We preliminarily selected 650nm band but other wavelengths could be equally useful”. Does it mean “We preliminarily selected 650nm band but other wavelengths within PAR spectral domain could be equally useful”?

Line 233: fPAR30 = 1 – SWr3. What’s the unit of SWr3? If it is W m-2, then this equation is wrong. Maybe, it should be fapar = 1 – albedo = 1 – SWr/SWin. With this method, fapar may be underestimated because Albedo_SW > Albedo_PAR, and also the seasonality in broadband albedo and par albedo may be different. Since seasonal APAR is mostly driven by the incoming PAR in clear sky condition, but it is better to mention these potential uncertainties.

Line 290-293: The same sentences are repeated.

Figure 3: I might have overlooked the description somewhere, but in Figure 3A  in the middle, there is a horizontal belt where a lot of Sif plots are concentrated (SiF ~0.005). What is this?

Section 3.2 and Figure 5: It’s nice to show NDVI-GPP and PRI-LUE relationships too. Then we can see how SiF could improve the potential uncertainties from other existing remote sensing indices.

Author Response

(The authors gave the same response as above.)
